# Effects of Muscle Type and Aging on Glycolysis and Physicochemical Quality Properties of *Bactrian camel* (*Camelus bactrianus*) Meat

**DOI:** 10.3390/ani14040611

**Published:** 2024-02-14

**Authors:** Haodi Lyu, Qin Na, Linlin Wang, Yafei Li, Zengtuo Zheng, Yinga Wu, Yuanyuan Li, Gai Hang, Xiangwei Zhu, Rimutu Ji, Fucheng Guo, Liang Ming

**Affiliations:** 1College of Food Science and Engineering, Inner Mongolia Agricultural University, Hohhot 010018, China; haodilv@163.com (H.L.); naqin_imau@163.com (Q.N.); 15203105978@163.com (L.W.); liyafei0903@163.com (Y.L.); zhengzengtuo@126.com (Z.Z.); 15648390406@163.com (Y.W.); m15335589563@163.com (Y.L.); 18548301027@163.com (G.H.); zhuxiangweiqiqi@163.com (X.Z.);; 2Inner Mongolia Institute of Camel Research, Alxa 737300, China; 3School of Life Science and Technology, Inner Mongolia University of Science and Technology, Baotou 014010, China

**Keywords:** *Bactrian camel*, muscle fiber type, glycolysis, aging, meat quality

## Abstract

**Simple Summary:**

Muscle fibers are the main component of skeletal muscle. Compared with cattle and sheep, camels have significantly thicker muscle fibers. We found that camel muscles with a high number of type I muscle fibers were more tender and muscles with a high content of type IIb muscle fibers were less tender, but more brightly colored, and the meat quality was greatly improved after aging. The quality of meat can be altered by choosing suitable feeding methods and aging times according to the different muscle fiber characteristics of camels, and also by choosing suitable processing methods according to the composition of muscle fibers in different parts of the muscle.

**Abstract:**

Poor tenderness of camel meat has seriously hampered the development of the camel meat industry. This study investigated the effects of muscle fiber composition and ageing time on meat quality, glycolytic potential, and glycolysis-related enzyme activities. Muscle samples of the longissimus thoracis (LT), psoas major (PM), and semitendinosus (ST) were collected from eight 8–10 year old Sonid *Bactrian camel*s (females). Muscle fiber composition was examined by ATPase staining and immunohistochemistry. Meat quality indexes, glycolytic potential, and activities of major glycolytic enzymes were examined at 4 °C aging for 1, 6, 24, 72, and 120 h. The results showed that LT was mainly composed of type IIb muscle fibers, whereas PM and ST were mainly composed of type I muscle fibers. The PCR results of the myosin heavy chain (MyHC) were consistent with the ATPase staining results. During aging, the shear force of LT muscle was always greater than that of PM and ST, and its glycolysis was the strongest; type IIa, IIb, and IIx muscle fibers were positively correlated with muscle shear force and glycolysis rate, and type I muscle fibers were significantly and negatively correlated with the activities of the key enzymes of glycolysis within 6 h. The results showed that the muscle fibers of LT muscle had the greatest glycolysis capacity. These results suggest that an excessive type IIb muscle fiber number percentage and area in camel meat accelerated the glycolysis process, but seriously affected the sensory profile of the camel meat. The results of this study provide directions for the camel industry when addressing the poor tenderness of camel meat.

## 1. Introduction

The Sonid *Bactrian camel* is mild-tempered and easy to be tamed, and has a strong ability to resist disasters and diseases. It is one of the high-quality camel breeds in China, and has been listed in China’s excellent livestock and poultry breeds resource list. Due to widespread misunderstanding of the flavor of camel meat, which is considered to be of poor quality, the camel industry has faced several noteworthy obstacles. Indeed, the level of tenderness exhibited by the meat from camels aged 1–3 years is comparable to that of beef [1]. Previous studies have indicated that camel meat shares similarities with other types of red meat and possesses exceptional physical sensory attributes [2,3]. With the exception of the relatively low lysine content in camel meat, the levels of other amino acids were found to be comparable to those seen in lamb meat [4]. Furthermore, it is worth noting that camel meat exhibits a substantial presence of fatty acids, vitamins, and minerals [5,6]. Consequently, it has garnered recognition as a nutritious alternative to high-fat and high-cholesterol meats, thus potentially serving as a novel addition to one’s daily dietary regimen. Nevertheless, according to the Food and Agriculture Organization, camel meat consumption accounts for only 0.18 percent of the world’s total red meat consumption and is concentrated in Muslim countries [5]. There are two primary factors contributing to the limited growth of the camel meat industry. Firstly, the existing sales chain for camel meat is imperfect, resulting in inefficiencies within the market. Additionally, camel meat products face a lack of competitiveness in the domestic market [7]. Secondly, consumer acceptance of camel meat is hindered by its thick muscle fibers, hardness, and challenging chewability. These factors collectively impede the rapid development of the camel meat industry.

After an animal is slaughtered, the muscle cells in its body rely primarily on glycolysis for energy metabolism. Glycolysis is a process in which glycogen (glucose, muscle glycogen, glucose 6-phosphate) is broken down into lactic acid, producing energy through the enzymatic reactions of lactate dehydrogenase (LDH), hexokinase (HK), glucose phosphorylase a (GPa), phosphofructokinase (PFK), and pyruvate kinase (PK) under anaerobic conditions [8]. The pH of muscle is reduced from 7.0 to approximately 5.7 by the accumulation of lactic acid [9]. The alteration in muscle pH, in terms of both rate and magnitude, has a direct impact on certain attributes associated with meat quality. Hence, understanding the changes in lactate content and pH during camel meat glycolysis can provide a basis for the production of high quality camel meat. A study found that meat that had been rapidly frozen to reduce the rate of decline in muscle pH had an *L** value of 57.86 after 96 h, which was significantly higher than the *L** value for meat that had been refrigerated at 4 °C (56.57) [10]. If there is an inadequate amount of glycogen in the muscle after slaughter, resulting in a high pH, an increase in the tethering hydrodynamic force, and overhydration of the myocytes, the muscle fibers will undergo a hardening process because of their tightly packed arrangement without contraction. This can therefore lead to poor meat tenderness [11]. The possibility of increased muscle glycogenolysis has a positive impact on reducing meat shear force and enhancing meat softness [12,13].

From a histological perspective, muscle fibers serve as the fundamental structural component of skeletal muscle. Furthermore, the quality of meat is significantly influenced by the composition and distinctive attributes of these muscle fibers [14]. Based on the different ATPase activities of muscle fibers in different pH environments, muscle fibers can be classified into slow oxidizing muscle fibers (type I), fast oxidizing muscle fibers (type IIa), and rapidly enzymatic muscle fibers (type IIb muscle fibers). Alternatively, based on myosin heavy chain (MyHC) isozymes (MYH7, MYH2, MYH4, and MYH1) they are classified as slow oxidizing muscle fibers (type I), fast oxidizing muscle fibers (type IIa), rapidly enzymatic muscle fibers (type IIb muscle fibers), and intermediate-type muscle fibers (type IIx) [15]. Muscle fibers have varying impacts on meat metabolism. In a study by Kim’s research group [16], the researchers observed that the longest muscle in the back of pigs exhibited a decline in type I and IIa muscle fibers as feeding time increased, while type IIb muscle fibers displayed an upward trajectory. Based on these findings, Kim conducted additional research and discovered that muscles with a high proportion of type IIb muscle fibers exhibited greater shear force and cooking loss compared with muscles with a low proportion of type IIb muscle fibers, when subjected to similar conditions. In addition, the cross-sectional area of muscle fibers, especially of type IIb muscle fibers, was found to be significantly and positively correlated with shear force [16]. According to Choe’s [17] study on muscle metabolism and muscle fiber types, type I muscle fibers exhibit relatively low levels of glycogen and lactate, but type II muscle fibers (IIa and IIb) demonstrate higher amounts of glycogen and lactate. A study also found that the calpain/calpain inhibitor ratio in type I muscle fibers was greater compared to oxidized muscle in type IIa and IIb muscle fibers. This results in the glycolytic enzymes of type I muscle fibers being able to maintain their activity for a longer period of time [18].

It has been found that ST has the significantly highest and lowest proportions of type I and type IIA muscle fibers, respectively, compared to other muscles [19]. The cross-sectional area of camel muscle fibers does not differ significantly between camel breeds, and glycolytic activity has been positively correlated with MyHC IIa [6]. It has also been found that the muscle characteristics of camels are affected by the seasons, with a high percentage of MyHC I in winter and MyHC IIa in fall [20]. To date, only a few studies have dealt with the effect of muscle fibers on camel meat quality and glycolysis during aging, and they have mainly focused on studies on dromedary camels.

In studies on dromedary camels, it has been found that the longissimus thoracis (LT) is dominated by type IIb muscle fibers and semitendinosus (ST) is dominated by type I muscle fibers [6], which facilitates the analysis of the effect of muscle fibers on camel meat quality and glycolysis. There have been no relevant studies on the muscle fibers of the camel psoas major (PM); however, studies on pigs, cows, and sheep have all found that the PM has a high degree of meat tenderness, which has a high economic value [21,22,23]. In China, camel meat is only a by-product, and camels are only considered as a source of protein after they have lost their economic value, such as for milk and fleece production, so camels destined for slaughterhouses tend to be older females [7]. For these reasons, in the present study, we used 8–10 year old Sonid *Bactrian camel* (female) to identify the muscle fiber types of the longissimus thoracis (LT), semitendinosus (ST), and psoas major (PM) muscles using ATPase staining and immunohistochemistry. We also evaluated the effect of muscle fiber type composition on glycolysis and changes in the meat quality of camels stored at 4 °C (1, 6, 24, 72, and 120 h).

The aim of this study was to evaluate the effect of muscle fiber type on the activity of the key enzymes of glycolysis, glycolytic potential, and overall meat quality, in order to provide a theoretical basis for solving the problem of camel carcasses with thick muscle fibers and poor eating quality, and to provide a theoretical basis for improving the quality of camel meat.

## 2. Materials and Methods

### 2.1. Materials

The study region encompassed the geographical coordinates 108°96′ to 112°95′ E and 42°88′ to 46°13′ N. This area was specifically located within a pasture situated in Sonid Right Banner, Xilingol League, China. The region experiences an annual mean temperature ranging from 0 to 3 °C, accompanied by a period of ice lasting up to 5 months and a cold period lasting up to 7 months. January has the lowest temperatures, characterized by an average of −20 °C.

The *Bactrian camel*s (*Camelus bactrianus*) under investigation were procured from the Sonid *Bactrian camel* Conservation Farm. A group of eight female Sonid *Bactrian camel*s that were in good health and aged 8 to 10 years were selected. These camels had an average weight of 794.50 ± 41.30 kg. The animal experimentation protocol was granted approval by the Animal Protection and Use Committee of Inner Mongolia Agricultural University (Permit No.2022-073) and was performed in accordance with the Chinese recommended standard (GB 14925-2010) using previously described methods [18]. All animals were reared in pastoral regions and exclusively consumed natural grass (mainly sun-dried alfalfa grass and antler grass) as their diet. All camels were subjected to identical environmental and management settings for a period of three months prior to slaughter, ensuring a consistent nutritional background. This animal study was thoroughly evaluated and the slaughter of camels followed ethical and fair principles.

The experimental camels underwent a period of fasting lasting 16 h before slaughter, while receiving unrestricted access to water throughout the duration of the study. The camels were slaughtered together at the Duolun county halal slaughterhouse, located in Inner Mongolia, China. The animals were slaughtered in accordance with halal food quality certification methods endorsed by the food industry, which adhere to the principles of Islam [19]. The temperature in the slaughterhouse was between 25 and 27 °C. During the slaughter procedure, the camels were subjected to exsanguination without prior stunning. Sampling was carried out in the slaughterhouse immediately after the workers had skinned the camels. The collection of all samples was completed within a 1 h time frame. The LT muscle was obtained from the 12th and 13th ribs, while the PM muscle was extracted from the midline of the vertebral column. The ST muscle was harvested from the thickest muscle located in the hind leg. The samples were cut into 1 cm × 1 cm × 3 cm strips after removal of fascia and fat, soaked in isopentane for 30 s to dehydrate, and then loaded into freezing tubes and stored in liquid nitrogen for transport back to the laboratory and then stored in a −80 °C refrigerator for ATPase staining of myofibers. Five grams of meat samples from each of LT, PM, and ST were stored in liquid nitrogen in freezing tubes and transported back to the laboratory, where they were also stored at −80 °C in a refrigerator for RNA extraction. Square samples of 500 g (excluding fascia and fat) were taken from each of the three muscle parts and stored in the refrigerator at 4 °C. The meat samples were tested for pH, color, cooking loss, and shear force at 1, 6, 24, 72, and 120 h of aging (meat quality indexes at 1 h were tested in the slaughterhouse, and those at 6, 24, 72, and 120 h were tested in the laboratory after being transported back to the Inner Mongolia Agricultural University). A 250 g sample of meat was taken from each muscle site and divided into 5 equal portions (shape not required) after removal of fascia and fat. The samples were stored in a refrigerator at 4 °C for 1, 6, 24, 72, and 120 h, and then stored in liquid nitrogen or at −80 °C in a refrigerator for testing the potential for glycolysis and the activity of the key enzymes of glycolysis.

### 2.2. Histological Analysis

Using a freezer slicer (Leica CM1520, Leica, Wetzlar, Germany), meat samples were sliced into 10 µm thick slices. ATPase staining of the camel meat was carried out with reference to Brooke’s method, with slight modifications [24]. The slides were placed in 100 mL staining vessels and 100 mL of 0.1 mol/L NaAc (acidic pre-incubation solution) was added, and the incubation process was completed by placing the slides on vibration for 5 min in a water bath at 37 °C. After the incubation, the staining vessel was removed and the waste liquid was discarded. An amount of 20 mL of 0.1 mol/L sodium barbiturate, 20 mL of 0.18 mol/L CaCl_2_, and 60 mL of distilled water (alkaline pre-incubation solution) were added to the staining vessel and the incubation was allowed to stand for 30 s at room temperature. At the end of the stationary period, the slides were transferred to a new water bath, and 20 mL of 0.1 mol/L NaAc, 10 mL of 0.18 mol/L CaCl_2_, and 60 mL of distilled water (incubation solution) were added, and the slides were allowed to stand for 50 min at 37 °C in a water bath. The pre-incubation solution was discarded and the sections were washed three times (for 3 min each) with 1% CaCl_2_ solution and then the sections were gently rinsed with 2% CoCl_2_ solution for 5 min. Using 0.01 mol/L C₈HN₂NaO₃, the staining vessel was rinsed (both sides, first 3 min, second 1 min). The staining vessels were rinsed continuously with distilled water for 1 min and stained with 100 mL of 1% (NH_4_)_2_S for 1 min.

Following a comprehensive washing procedure with water, the slides were subsequently sealed with neutral gum, enabling the subsequent observation of the morphology of the muscle tissue [24]. Images of the muscle fibers were captured at a magnification of 100× using a light microscope (Leica DMI4000B, Leica, Wetzlar, Germany). The muscle fiber characteristics were analyzed using Leica Qwin V3 software for image processing, using the data measured for each relevant contour. The overall count of muscle fibers exceeded 200, and the selection of muscle fibers to be measured was conducted in a randomized manner, taking into account the absence of tissue damage and freezing injury.

### 2.3. Real-Time Quantitative PCR of MyHC Gene Isoforms

From the −80 °C refrigerator, 20 mg of meat samples were taken and homogenized using an electric homogenizer (F10, FLUKO, Shanghai, China). RNA was extracted using a Total RNA Isolation Kit (Thermo Fisher Scientific, Waltham, MA, USA). The concentration of the extracted RNA was greater than 300 ug/mL. The RNA was then reverse transcribed to cDNA using the Takara PrimeScript RT kit (Takara, Dalian, China). *GAPDH* (encoding glyceraldehyde-3-phosphate dehydrogenase) was detected as an internal reference. Primers for the target genes (*MYH7*, *MYH2*, *MYH4*, and *MYH1*) were designed with the help of Primer Premier 5.0 software (PREMIER Biosoft International, Palo Alto, CA, USA) and synthesized by Invitrogen Technologies (Shanghai, China). The qPCR step of the qRT-PCR procedure was achieved by fluorescence staining of the cDNA using Takara’s Terra qPCR Direct TB Green ^®^ Premix kit, and a LightCycler96 fluorescence quantitative PCR instrument (ROCHE, Indianapolis, IN, USA). Reagent configurations for qPCR are shown in the Appendix A. The primer sequences are shown in Table 1.

The thermal cycling program was as follows: firstly, 30 s at 95 °C, followed by 40 cycles of denaturation at 95 °C for 5 s, annealing at 57 °C for 30 s, and finally extension at 72 °C for 30 s. All the experimental sample analyses were run in triplicate. The gene expression data were calculated using the 2^−∆∆Ct^ method [25].

### 2.4. Detection of Camel Meat Quality Indicators

#### 2.4.1. pH

The pH was measured using the spot method, where 10 g of meat sample was taken from each site and tested using a pH meter (FE28 benchtop; Mettler Toledo, Columbus, OH, USA) on muscle sites with different times of acid excretion. Three measurements were taken for each sample and the mean recorded.

#### 2.4.2. Color

Meat samples were removed from the 4 °C refrigerator and cut into 5 cm × 5 cm × 5 cm shapes. The objective color values brightness (*L**), redness (*a**), and yellowness (*b**) of the meat samples were measured using a Minolta CR-5 colorimeter TC-P2A automatic colorimeter (Shanghai Biological and Biochemical Experimental Instrument Co., Ltd., Shanghai, China). The same sample was measured three times, and the average value was calculated.

#### 2.4.3. Cooking Loss

For LT, PM, and ST, 100 g of meat each was taken and placed in a water bath preheated to 75 °C. The center temperature of the meat was monitored using a digital display thermometer (Shanghai Automatic Instrument Co., Ltd., Shanghai, China). When the center temperature reached 70 °C, the sample was removed, placed at room temperature, and the surface moisture was gently wiped off with absorbent paper. The sample was weighed and recorded as *m*_1_.
(1)$$\mathrm{Cooking}\;\mathrm{loss}\;\%\;=\frac{100-m_{1}}{100}\times 100$$

#### 2.4.4. Shear Force

About 200 g of each meat sample was placed into a preheated 85 °C water bath and the center temperature of the meat was monitored using a digital display thermometer. When the center of the meat reached a temperature of 70 °C, samples were removed and dried with absorbent paper to indicate moisture. Along the direction of the muscle fibers, the meat samples were cut into 1 × 3 × 1 cm rectangles. Shear force was measured using a C-LM3B digital tenderness meter (C-LM, College of Engineering, Northeastern Agricultural University, Beijing, China) and the results were expressed in Newtons (N). The same muscle part was tested 3 times and the average value was taken.

### 2.5. Glycolytic Potential and Key Enzyme Activity

#### 2.5.1. Glycolytic Potential (GP)

The GP indicators, such as glucose, muscle glycogen, glucose-6-phosphate (6PG), and lactic acid (LA) were detected using a glucose detection kit, a muscle glycogen kit, a lactic acid kit, and a 6PG kit (Nanjing Jiancheng Bioengineering Institute, Nanjing, China), respectively. The GP was calculated according to the formula: GP = 2(glucose + muscle glycogen + 6PG) + LA.

#### 2.5.2. Key Enzyme Activity

The activities of LDH, HK, GPa, PFK, and PK in the muscle were determined using an assay kit (SOLEIBO Biotechnology Co., Ltd., Beijing, China). The operation was performed according to the kit instructions, and the activities of the different enzymes were calculated according to the formula provided by the kit.

### 2.6. Data Analysis

The data were systematically arranged and subjected to statistical analysis using Microsoft Excel 2010 software (Microsoft Corp., Redmond, WA, USA). The results are presented as the mean ± the standard deviation. The results were analyzed by one-way analysis of variance (ANOVA) using SPSS 2007 software. Means were compared using the least significant difference (LSD) procedure. *p* < 0.05 was considered statistically significant. The correlation coefficients of muscle fiber and ageing time on meat quality and glycolysis were assessed using Pearson Correlation Analysis (SPSS 27 Bivariate Correlation Analysis). Origin 2021 (OriginLab Corp., Northampton, MA, USA) was employed to create visual representations of the data.

## 3. Results

### 3.1. Types of Muscle Fibers in Different Camel Muscles

#### 3.1.1. ATPase Staining

The results of muscle fiber type staining in the Sonid *Bactrian camel* are shown in Figure 1. Black represents type I muscle fibers, bright gray represents type IIb muscle fibers, and dark gray represents type IIa muscle fibers.

The number, composition, and area of muscle fibers in the different muscles are shown in Table 2. The LT muscle was dominated by type IIb fibers, accounting for 36.14 ± 1.38%, and the type IIa fibers of LT accounted for 34.36 ± 1.95%, which was significantly higher than that in the PM and ST muscles (*p* < 0.05). The PM and ST muscles were mainly composed of type I muscle fibers, accounting for 45.67 ± 1.34% and 40.69 ± 0.67%, respectively. In the same muscle, the proportion of type I muscle fibers was significantly higher than the proportion of type Ⅱa and IIb muscle fibers in the PM and ST muscles (*p* < 0.05). Among the different muscles, the proportion of type IIa fibers in LT (34.36%) was significantly higher than that in PM (27.59%) and ST (4.55%) (*p* < 0.05), while the proportion of type I in LT (29.50%) was significantly lower than that in PM (45.67%) and ST (40.69%) (*p* < 0.05). In the three muscles, the area of type IIb fibers was significantly larger than the areas of type IIa and type I fibers. The area of type IIa muscle fibers in LT and ST were significantly larger than those of type I fibers.

#### 3.1.2. Expression of MyHC Isoform mRNAs in the Different Muscles of Camel

The results for MyHC mRNA expression in the different muscles of camel are shown in Table 3. The expression levels of MyHC I, IIa, and IIb mRNAs were basically consistent with the proportion of muscle fibers of type I, IIa, and IIb in the ATPase staining results. The MYH7Ix mRNA expression level in the LT muscle was significantly higher than that of MyHC I, IIa, and IIb mRNAs (*p* < 0.05), while the highest expression of MyHC I mRNA was found in the PM and ST muscles (*p* < 0.05). The MYH7Ix mRNA expression in the LT muscle was significantly higher than that in the PM and ST muscles, and MyHC I mRNA expression in the PM and ST muscles was significantly higher than that in the LT muscle (*p* < 0.05). The expression of MyHC IIb mRNA in the ST muscle was significantly higher than that in the LT and PM muscles (*p* < 0.05).

### 3.2. Quality of Camel Meat

#### 3.2.1. pH

The results of pH changes in different muscles of camel meat during aging are shown in Table 4. Lactic acid is produced by glycolysis after slaughter [7]; therefore, the pH of all three muscles showed a downward trend during the aging progress. Compared with that at 1 h, the pH value of the LT muscle showed a significant decrease at 6 h (*p* < 0.05), after which there was no significant change during aging from 6 h to 120 h. At 6 h and 120 h, the pH values of the LT muscle were 5.76 ± 0.04 and 5.62 ± 0.06, respectively, which were significantly lower than those of the PM and ST muscles (*p* < 0.05) at the same aging times. The results showed that the process of glycolytic lactic acid production in LT muscle mainly occurred within 6 h after slaughter, and the lactic acid accumulation rate of LT was higher than that of PM and ST.

#### 3.2.2. Shear Force

The changes in shear force for the different muscles of camel during aging are shown in Table 5. The shear force of the LT, PM, and ST muscles reached a maximum at 24 h and showed a significant decreasing trend at 72 h compared with that at 24 h (*p* < 0.05), indicating that camel meat entered the self-contained stage at 72 h of aging, and the tenderness of meat was improved. Between different parts, we also found that the shear force of LT was always significantly higher than that of PM and ST (*p* < 0.05), while the shear force of PM at 120 h was significantly lower than that of LT and ST (*p* < 0.05).

#### 3.2.3. Cooking Loss

The changes of cooking loss in the different muscles of camel during aging are shown in Table 6. The cooking loss of the three muscles showed a trend of an initial decrease and then an increase during aging. The cooking losses of LT and ST reached their maximum in 72 h, but were decreased significantly at 120 h compared with those at 72 h (*p* < 0.05). The cooking loss of PM reached a maximum at 24 h, and then decreased at 72 h and 120 h compared with that at 24 h.

#### 3.2.4. Color

As shown in Table 7, with prolonged aging time, the *L** and *b** values displayed a continuous upward trend and the *a** value increased first and then decreased in the LT, PM, and ST muscles. Among different camel muscles, the *L** value of LT was always significantly higher than that of PM and ST (*p* < 0.05), and the *a** and *b** values of PM were significantly higher than those of LT at 1–120 h (*p* < 0.05).

### 3.3. Glycolytic Potential

We investigated the indicators related to the GP of different muscles of camel meat (Table 8). The glucose, muscle glycogen, and 6GP content of LT remained the highest at 1 h and 6 h. At 120 h, ST had the highest levels of glucose, muscle glycogen, and 6PG. The lactate content of LT was consistently higher than that of PM and ST. These data indicated that the conversion rates of glucose, muscle glycogen, and 6PG in the LT muscle are higher than those in PM and ST. It is worth noting that the trend of lactic acid is the same as that of the pH, and the lactic acid content of different muscles is inversely proportional to the pH, suggesting that the main cause of the change in pH in the samples in this experiment was the content of lactic acid.

### 3.4. Enzyme Activity

The changes of glycolytic enzyme activities in the three muscles are shown in Figure 2. The GPa activity in the three muscles was significantly increased at 6 h compared with that at 1 h (*p* < 0.05), and was significantly decreased at 24 h compared with that at 6 h (*p* < 0.05), which was consistent with the trend of glucose changes. This indicated that the breakdown of glucose during glycolysis occurred mainly in the first 6 h, which was consistent with the conclusions drawn from detecting changes in pH and lactic acid. The GPa activity of LT was significantly higher than that of PM and ST at 6, 72, and 120 h (*p* < 0.05).

The PFK activity of the three camel muscles showed a decreasing trend during aging, with LT, PM, and ST showing a significant decrease in PFK activity at 6 h compared with that at 1 h (*p* < 0.05), and also at 72 h compared with that at 24 h (*p* < 0.05). The lowest PFK activity was found in PM and the highest in LT at 24 h.

The activity of HK showed a significant increase at 6 h, but there was no significant difference at 1, 24, 72, and 120 h. Among the different muscles, only at 6 h was the HK activity of the LT muscle (175.13 U/mg of protein) significantly higher than that of the ST (154.39 U/mg of protein) and PM (153.49 U/mg of protein) muscles.

The LDH activity of all muscles showed a decreasing trend from 1 to 24 h, after which it stabilized. The LDH activity of LT (4.77 U/mg of protein) was significantly higher than that of PM (4.27 U/mg of protein) and ST (4.39 U/mg of protein) at 1 h (*p* < 0.05); however, there was no significant difference between the different muscles at other aging times.

PK is the last rate-limiting enzyme of the glycolytic pathway, and the PK activity of the three muscles showed an increasing and then decreasing trend. At 6 h, the PK activity of LT (4.71 U/mg of protein) was significantly higher than that of PM (3.22 U/mg of protein) and ST (2.81 U/mg of protein). At 24 h and 72 h, the PK enzyme activity of LT (4.16 U/mg of protein and 3.84 U/mg of protein) was significantly lower than that of PM (4.65 U/mg of protein and 4.55 U/mg of protein) and ST (4.89 U/mg of protein and 4.13 U/mg of protein) (*p* < 0.05). This indicated that the rate of glycolysis in LT was higher than that in PM and ST.

### 3.5. Relationship between Meat Quality and Glycolytic Indices and Muscle Fiber Type

Correlation coefficients (r) of meat quality with different proportions of muscle fiber types, different muscles fiber type areas, and *MyHC* gene subtype expression during camel meat aging are shown in Figure 3. *MyHCⅠ* expression correlated with the proportion of type Ⅰ fibers, *MYH2* expression correlated with the proportion of type Ⅱa fibers, and *MYH4* expression correlated with the proportion of type Ⅱb fibers (*p* < 0.05). Notably, *MYH7Ix* expression consistently correlated with the proportion of type Ⅱb fibers for many indexes.

pH correlated significantly and positively with the proportion of type I fibers at 1, 6, and 120 h, and with *MYH7* expression at 1 and 120 h. This indicated that the higher the proportion of type I muscle fibers and the expression of *MYH7*, the higher the pH. There was also a significant negative correlation between pH and the proportion of type IIa fibers at 6 h, type IIb fibers at 1 h, and MYH7Ix expression at 1 and 120 h. This indicated that the higher the proportion of type IIa and IIb muscle fibers, and the expression level of *MYH7Ix*, the more the pH decreased during maturation.

There was a significant negative correlation between the shear force and the proportion of type I fibers at 1–120 h (*p* < 0.05), and a significant negative correlation with *MYH7* expression at 1, 6, 72, and 120 h (*p* < 0.05). These results indicated that the number of type I muscle fibers and MYH7 expression have a negative impact on the shear force during camel meat aging. The proportion of type IIb muscle fibers correlated significantly and negatively with the shear force at 1 and 120 h (*p* < 0.05), and the cross-sectional area of type IIb fibers correlated significantly and negatively with the shear force at 24, 72, and 120 h (*p* < 0.05), indicating that the greater the number of type IIb fibers, the larger the cross-sectional area, and the greater the shear force.

Cooking loss correlated significantly and negatively with the proportion of type IIa fibers at 6 h; however, at the other time points, the correlation was inconsistent and there was no clear pattern.

The *L** values showed highly significant and negative correlations with both the proportion of type I fibers and *MYH7* expression from 1 to 120 h, significant positive correlations with type IIa and IIb fibers from 1 to 120 h, significant positive correlations with *MYH7Ib* expression from 1 to 72 h, and highly significant positive correlations with *MYH7Ix* expression from 1 to 120 h (*p* < 0.05). The *a** values were significantly and positively correlated with the proportion of type I fibers at 1 to 6 h and *MyHC* expression at 1 to 72 h (*p* < 0.05), and significantly and negatively correlated with the proportion of type IIb fibers at 1 to 120 h, MyHC IIb expression at 72 h and *MYH7Ix* expression at 1 to 72 h (*p* < 0.05). The *b** values were significantly and positively correlated with the proportion of type I fibers and *MYH7* expression at 6 to 120 h (*p* < 0.05).

The correlation coefficients (r) of different muscle fibers proportions, areas, and *MyHC* gene expression levels with the GP and key enzyme activities during camel carcass aging are shown in Figure 4. Glucose levels correlated positively with the proportion of type IIb fibers at 1–72 h and with *MYH7Ix* expression at 1 and 72 h (*p* < 0.05), and correlated significantly and negatively (*p* < 0.05) with the proportion of type I fibers at 1 h and with *MYH7* expression at 1, 6, and 72 h (*p* < 0.05). The amount of muscle glycogen correlated significantly and positively with the proportion of type IIb fibers and the expression levels of *MYH7Ib* and *MYH7Ix* at 1 h (*p* < 0.05). Lactate levels were highly negatively correlated with the number of type I muscle fibers and with *MYH7* expression at 1–120 h (*p* < 0.05), significantly and positively correlated with type IIb at 1–6 h (*p* < 0.05), and highly significantly and positively correlated with type IIb at 24–120 h (*p* < 0.01). Notably *MYH7Ix* expression was consistently and highly significantly positively correlated with lactate levels at 1–120 h (*p* < 0.01). GP was highly significantly negatively correlated with the proportion of type I fibers at 1, 6, 72, and 120 h, with the proportion of type IIb fibers and *MYH7Ix* expression from 1 to 120 h (*p* < 0.01), and was significantly and positively correlated with *MYH4* expression at 1, 6, 72, and 120 h (*p* < 0.05).

The correlation between other glycolytic enzyme activities and muscle fiber types was not significant at 24 to 120 h; however, the proportion of type I muscle fibers was found to be always negatively correlated with enzyme activity at 1 and 6 h, and the proportion number of type IIb muscle fibers was positively correlated with enzyme activity. The above results indicated that at 1–6 h, the greater the number of type IIa muscle fibers, the slower the rate of glycolysis, and the greater the number of type IIb muscle fibers, the faster the rate of glycolysis.

## 4. Discussion

The composition of muscle fiber types is affected by species, exercise, and nutrient intake [26,27,28]. Kadim [29] studied the muscle fibers of different muscles of 1.5-year-old single-humped camels and found that the LT muscle was mainly composed of type Ⅱb muscle fibers, accounting for 35.36 ± 3.10%, and the ST muscle was mainly composed of type I muscle fibers, which is consistent with the experimental results of the present study. The area of muscle fibers in different muscles were type Ⅱb > type IIa > type I, which is consistent with previous research results [14,16,29]. Maltin [30] and Ruusunen [31] found that the more motor functions a muscle undertakes, the larger the area of the muscle fibers that make up the muscle. This finding could explain the significantly smaller area of PM muscle type IIa fibers compared with that in the LT and ST muscles observed in the present study (*p* < 0.05). Allen [32] found that as animals age, the MyHC subtype tended to change from I→IIa→IIx, and the cross-sectional area of the muscle also became significantly reduced. Comparison of our results with those of Kadim’s [29] revealed that the areas of the different types of muscle fibers in 8–10 year old camels were smaller than those of 1–3 year old camels, and the number of type IIa muscle fibers was greater than that of 1–3 year old camels. There have been no previous studies on the expression of *MyHC* isozyme genes in camels; therefore, it cannot be determined that there is a change trend of MyHC isoforms from I→IIa→IIx in camel meat. A comparison of the muscle fiber areas between the different species reveals that the areas of the different types of muscle fibers in camel LT muscle are 3365.31 µm^2^ (type I muscle fibers), 5521.98 µm^2^ (type IIa muscle fibers), and 7520.45 µm^2^ (type IIb muscle fibers), respectively. This is much higher than 1019.59 µm^2^, 1574.39 µm^2^, and 1242.41 µm^2^ in sheep LT, which seems to be the main reason why the shear force of camel meat is much higher than that of other species [33].

The type of muscle fiber is the main factor affecting the quality change during meat maturation [34,35,36]. Kim believed that the increase in the proportion of type IIb muscle fibers was related to the decrease in water holding capacity [16]. Mookerjee found that the content of metmyoglobin and heme iron in type I muscle fibers were higher and the color was redder [7]. In addition, a study showed a positive correlation between the surface of type IIb muscle fibers and shear force, with a faster glycolysis rate [37]. Similar findings were reported in the present study. In addition, the proportion of type IIb muscle fibers in LT was significantly lower than that in PM (*p* < 0.05), and the pH threshold in PM was also significantly lower than that in LT (*p* < 0.05). There was a negative correlation between type IIb muscle fibers and pH threshold. Kadim’s [16] research on muscle fibers found that the cross-sectional area of muscle fibers correlated positively with the shear force, and the cross-sectional area of type IIb muscle fibers was significantly larger than that of type I muscle fibers. Similarly, in this study, type IIb (40.67 ± 2.03%) dominated the LT muscle, and its shear force was significantly larger than that of PM and ST. In the correlation between muscle fiber type and meat quality, we found that type IIb fibers correlated positively with shear force at 1 h and 120 h (*p* < 0.05). In addition, the results also demonstrated that type I fibers had a very significant negative correlation with shear force. No significant correlation was found between cooking loss and muscle fiber type. The change rule of meat color was consistent with the change rule of camel meat reported by Maqsood [38]. The *a** and *b** values of LT were lower than those of PM and ST. This is because the proportion of type I muscle fibers in LT is significantly lower than that in PM and ST (Table 3). Type I muscle fibers have been shown to have higher heme and myoglobin content, especially in camel meat compared with that in other species [38,39]. The *L** value of LT was always higher than that of PM and ST, which is related to the fact that LT mainly comprises type IIb fibers. Type IIb fibers have a strong glycolysis ability and thus exude more water [37].

After slaughter, the oxygen supply to muscle tissues is interrupted and the energy metabolism of muscle cells is dominated by anaerobic respiration. Muscle glycogen, glucose, and 6PG produce lactic acid through the glycolytic pathway, resulting in a decrease in pH [7]. The trends of pH, muscle glycogen, glucose, 6PG, and lactic acid in the three muscles of the present experiment were consistent with the above theory. Ryu’s research group [40] and Plastow’s research group [41] all found that the glycolytic enzyme activities and glycogen contents of type IIb muscle fibers were higher than those of type I muscle fibers, this is consistent with the findings of the present study that PM and ST (composed mainly of type I muscle fibers) have lower glycolytic enzyme activity and glycolytic potential than LT (composed mainly of type IIb muscle fibers). The relationship between the magnitude of glycolytic enzyme activity and glycogen content of PM and ST in the present study was similar to that reported by Ryu and Plastow. Bai [42] suggested that type I muscle fibers inhibit glycolysis of muscle glycogen and slow down the decrease in pH value during muscle aging. In the present study, an inhibitory effect of type I muscle fibers on glycolysis was found only within 6 h. After 6 h, the effect of type I muscle fibers on glycolytic enzyme activity did not show a clear regularity.

## 5. Conclusions

The results of this study showed that the composition of muscle fiber types varied in different muscle parts. Different muscle fiber types also have different effects on changes in meat quality and glycolysis during maturation. The higher the percentage of type I muscle fiber and the percentage of MyHC I expression in the muscle, the lower the shear force of the meat and the better its tenderness in the early stages of aging. The higher the proportion of type IIa and type IIb muscle fibers, the higher the fiber area, and the higher the expression of MyHC IIb and MyHC IIx, the higher the shear force and the poorer the tenderness of the meat at the beginning of aging. However, with prolonged aging times, samples with higher ratios of IIa and IIb muscle fibers and higher expression of MyHC genotypes IIa, IIb, and IIx showed greater improvement in meat quality. In addition, type I fibers and MyHC I expression had an inhibitory effect on glycolysis, whereas type IIa and type IIb fibers, and MyHC IIa, MyHC IIb, and MyHC IIx expression, promoted glycolysis at 6 h. This study investigated the muscle fiber composition of different parts of camel meat and its effect on meat quality and glycolysis during aging, providing theoretical support for the fine processing of camel meat by parts.

## Figures and Tables

**Figure 1 animals-14-00611-f001:**
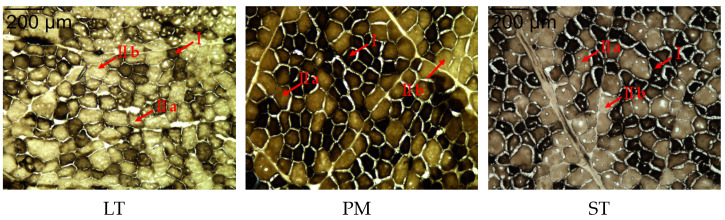
Results of ATPase staining of LT, PM, and ST in Sonid *Bactrian camel*. Magnification of 200×. I: fiber type I; IIa: fiber type IIa; IIb: fiber type IIb.

**Figure 2 animals-14-00611-f002:**
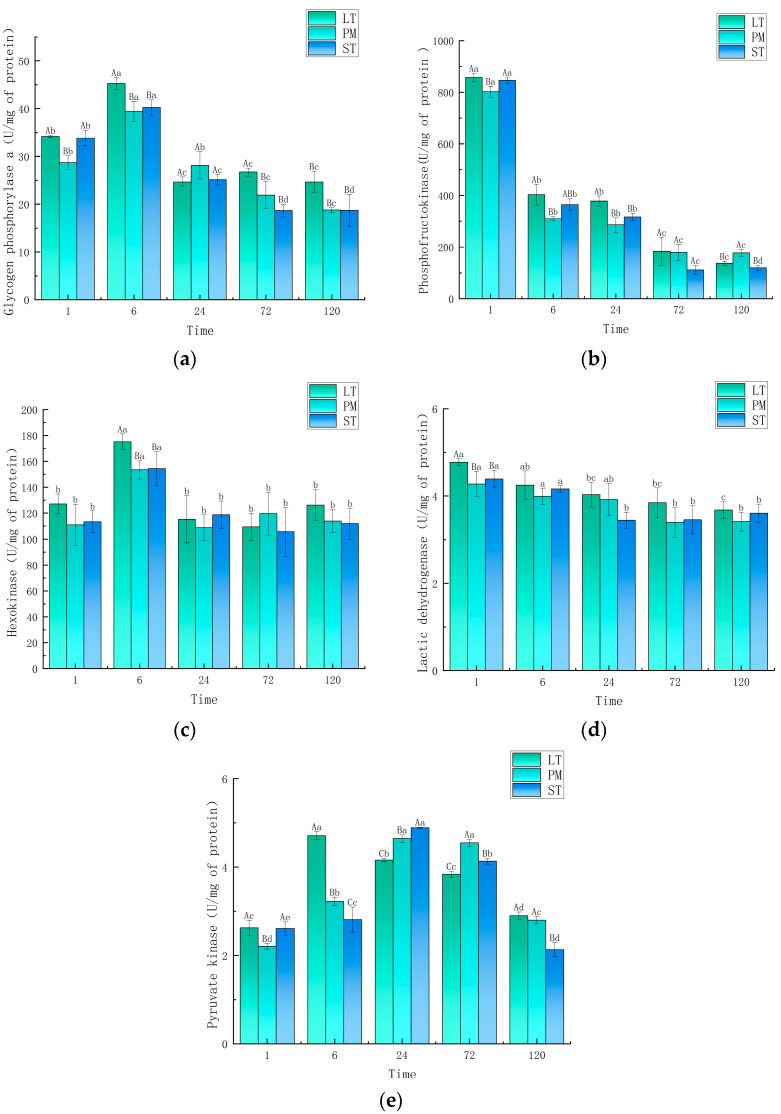
Changes in activity of key enzymes of glycolysis: (**a**) glucose phosphorylase a, (**b**) phosphofructokinase, (**c**) hexokinase, (**d**) lactic acid dehydrogenase, and (**e**) pyruvate kinase. Different capital letters indicate significant differences between groups (*p* < 0.05), and different lowercase letters indicate significant differences within groups (*p* < 0.05). LT, longissimus thoracis muscle; PM, psoas major muscle; ST, semitendinosus (ST) muscle.

**Figure 3 animals-14-00611-f003:**
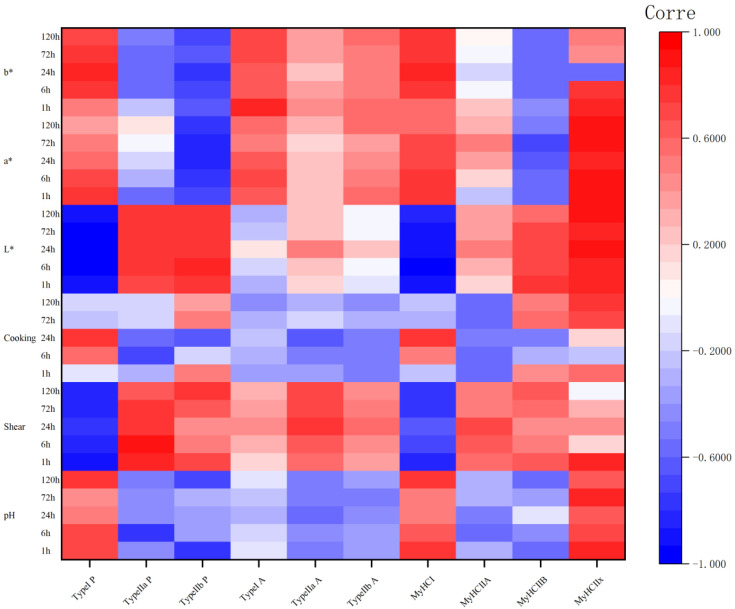
The correlation coefficient (r) heatmap between the proportion and area of different muscle fiber types, as well as *MyHC* gene subtype expression levels, and meat quality during the aging process of camel meat. P: proportion of muscle fiber types; A: area of muscle fiber types. Red indicates positive correlation and blue indicates negative correlation.

**Figure 4 animals-14-00611-f004:**
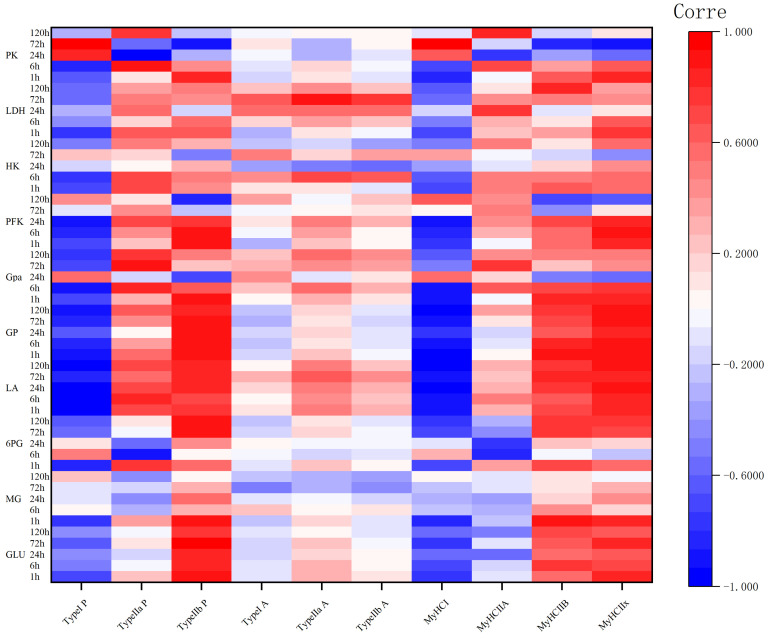
Correlation coefficient (r) heatmap between the percentage and area of muscle fiber types, and *MyHC* gene subtype expression levels of different muscle fiber types during camel meat aging, glycolytic potential, and key enzyme activity. LDH, lactate dehydrogenase; HK, hexokinase; GPa, glucose phosphorylase a; PFK, phosphofructokinase; PK, pyruvate kinase; GP, glycolytic potential; LA, lactic acid; 6PG, glucose-6-phosphate; MG, muscle glycogen; GLU, glucose. Red indicates positive correlation and blue indicates negative correlation.

**Table 1 animals-14-00611-t001:** Primers used for quantitative real-time PCR analysis.

Genes	Primer Sequences (5′→3′)	Product Size, bp	Accession NO.
*MYH7*	F: CTGTCCAAGTTCCGCAAGGTG R: TGGCAAATCTACTCCTCATTCAAGC	143	XM_032481727.1
*MYH2*	F: AAGAACATGGAACAGACGGTGAAG R: TGCTCACTCTCAACCTCTCCTTC	159	XM_032498795.1
*MYH4*	F: CGACATTGACCACACCCAGTAC R: GCTTTTCATCTCGCATCTCCTCTAG	95	XM_032498791.1
*MYH1*	F: TGCCAAATGTGCTTCCCTTGAG R: TTTCCATTCTGCTAGGACCTTATCG	148	XM_032498792.1
*GAPDH*	F: CTCTGGCAAAGTGGACATTGTTG R: TGGGTGGAATCATACTGGAACATG	87	XM_032472950.1

**Table 2 animals-14-00611-t002:** Muscle fiber type proportion and area in LT, ST, and PM of Sonid *Bactrian camel*s.

Measurement	Muscle		
	LT	PM	ST
Proportion (%)			
Type I	29.50 ± 1.06 ^Ca^	45.67 ± 1.34 ^Aa^	40.69 ± 0.67 ^Ba^
Type IIA	34.36 ± 1.95 ^Ab^	27.59 ± 1.44 ^Bb^	24.55 ± 0.23 ^Bc^
Type IIB	36.14 ± 1.38 ^Ab^	26.74 ± 0.76 ^Bb^	34.76 ± 0.56 ^Ab^
Area (μm^2^)			
Type I	3365.31 ± 530.23 ^Ac^	3409.66 ± 449.23 ^Ab^	3649.93 ± 664.54 ^Ac^
Type IIA	5521.98 ± 379.66 ^Ab^	3814.32 ± 565.07 ^Cb^	4480.07 ± 444.73 ^Bb^
Type IIB	7520.45 ± 1049.13 ^Aa^	6623.99 ± 869.38 ^Ba^	6966.12 ± 874.18 ^ABa^

Uppercase letters indicate significant differences between muscles (*p* < 0.05) and lowercase letters indicate significant differences between muscle fiber types (*p* < 0.05).

**Table 3 animals-14-00611-t003:** *MyHC* gene isoforms expression levels in various muscles.

Genes	LT	PM	ST
*MYH7*	13.42 ± 2.01 ^Cb^	63.90 ± 9.95 ^Aa^	37.83 ± 8.7 ^Ba^
*MYH2*	18.12 ± 2.68 ^Ab^	15.91 ± 1.66 ^ABb^	11.30 ± 2.39 ^Bb^
*MYH4*	31.48 ± 8.72 ^Aa^	15.44 ± 2.38 ^Bb^	28.04 ± 5.22 ^Aab^
*MYH1*	36.98 ± 8.77 ^Aa^	4.75 ± 2.14 ^Cb^	22.83 ± 3.26 ^Bab^

Data in the columns are the relative mRNA expression levels in the three muscle types. Uppercase letters indicate significant differences between muscles (*p* < 0.05) and lowercase letters indicate significant differences between muscle fiber types (*p* < 0.05).

**Table 4 animals-14-00611-t004:** Changes in pH during aging of different muscles of camel.

Time (h)	pH (LT)	pH (PM)	pH (ST)
1	6.01 ± 0.13 ^Ba^	6.31 ± 0.03 ^Aa^	6.15 ± 0.18 ^Aa^
6	5.76 ± 0.04 ^Bb^	5.97 ± 0.11 ^Ab^	5.98 ± 0.12 ^Aab^
24	5.79 ± 0.10 ^Ab^	5.87 ± 0.06 ^Abc^	5.88 ± 0.05 ^Abc^
72	5.70 ± 0.04 ^Ab^	5.79 ± 0.07 ^Ac^	5.77 ± 0.10 ^Abc^
120	5.64 ± 0.06 ^Bb^	5.74 ± 0.03 ^Ac^	5.68 ± 0.06 ^ABc^

Uppercase letters indicate significant differences between muscles (*p* < 0.05) and lowercase letters indicate significant differences between times (*p* < 0.05).

**Table 5 animals-14-00611-t005:** Changes in shear force (N) during aging of different muscles of camel meat.

Time (h)	LT	PM	ST
1	79.15 ± 3.68 ^Ab^	64.13 ± 2.74 ^Bc^	67.33 ± 2.64 ^Bb^
6	85.14 ± 4.95 ^Aa^	76.80 ± 1.71 ^Bb^	75.19 ± 3.78 ^Ba^
24	87.41 ± 2.85 ^Aa^	79.54 ± 2.21 ^Ba^	78.14 ± 1.86 ^Ba^
72	73.56 ± 2.71 ^Ab^	64.19 ± 3.31 ^Bd^	64.32 ± 0.75 ^Bc^
120	70.25 ± 4.93 ^Ac^	60.85 ± 1.69 ^Ce^	63.65 ± 2.28 ^Bc^

Data in the columns represent the shear force in the three muscle types. Uppercase letters indicate significant differences between muscles (*p* < 0.05) and lowercase letters indicate significant differences between times (*p* < 0.05).

**Table 6 animals-14-00611-t006:** Changes in cooking losses (%) during aging of different muscles of camel.

Time (h)	LT	PM	ST
1	29.85 ± 1.29 ^Ac^	28.87 ± 1.38 ^Ab^	31.55 ± 1.30 ^Ac^
6	28.66 ± 0.77 ^Ac^	31.04 ± 1.27 ^Ab^	31.83 ± 3.16 ^Ac^
24	31.87 ± 3.11 ^Bbc^	37.34 ± 1.41 ^Aa^	35.74 ± 2.12 ^ABb^
72	38.24 ± 2.83 ^ABa^	36.28 ± 1.17 ^Ba^	40.64 ± 1.62 ^Aa^
120	36.04 ± 3.50 ^ABab^	34.47 ± 2.96 ^Ba^	38.60 ± 1.13 ^Aab^

Data in the columns represent the cooking losses in the three muscle types. Uppercase letters indicate significant differences between muscles (*p* < 0.05) and lowercase letters indicate significant differences between times (*p* < 0.05).

**Table 7 animals-14-00611-t007:** Changes in color values during the aging of different muscles of camel.

Color	Muscle Type	LT	PM	ST
*L**	1 h	30.78 ± 1.03 ^Ab^	25.56 ± 1.01 ^Bb^	27.64 ± 0.31 ^Bc^
	6 h	31.57 ± 0.84 ^Aab^	26.47 ± 2.40 ^Bab^	28.23 ± 0.32 ^Bb^
	24 h	32.70 ± 2.50 ^Aa^	27.57 ± 1.29 ^Ca^	28.99 ± 0.18 ^Ba^
	72 h	32.22 ± 0.66 ^Aa^	28.10 ± 0.09 ^Ba^	29.23 ± 0.39 ^Bd^
	120 h	32.08 ± 0.77 ^Aa^	28.03 ± 1.62 ^Ca^	29.01 ± 0.04 ^Bb^
*a**	1 h	9.67 ± 0.65 ^Ba^	13.67 ± 1.19 ^Aa^	12.11 ± 1.62 ^Aa^
	6 h	12.02 ± 0.90 ^Bb^	16.23 ± 2.26 ^Ab^	13.16 ± 0.36 ^Ba^
	24 h	8.84 ± 1.00 ^Bb^	12.99 ± 2.15 ^Ab^	8.86 ± 0.11 ^Bb^
	72 h	9.01 ± 1.37 ^Bb^	13.06 ± 0.89 ^Ab^	7.94 ± 0.12 ^Bb^
	120 h	8.38 ± 0.10 ^Bb^	10.20 ± 0.79 ^Ac^	7.55 ± 1.03 ^Bb^
*b**	1 h	4.24 ± 0.45 ^Bb^	5.63 ± 1.16 ^Ab^	4.60 ± 0.10 ^ABb^
	6 h	4.96 ± 0.83 ^Cab^	7.34 ± 0.70 ^Aa^	6.19 ± 0.31 ^Ba^
	24 h	5.37 ± 0.33 ^Ba^	7.26 ± 0.65 ^Aa^	6.42 ± 0.52 ^Ba^
	72 h	5.66 ± 1.01 ^Ca^	7.50 ± 0.76 ^Aa^	6.70 ± 0.19 ^Ba^
	120 h	5.84 ± 0.75 ^Ca^	8.18 ± 1.13 ^Aa^	6.93 ± 0.27 ^Ba^

*L** (luminance axis) indicates black and white, 0 is black, 100 is white, and between 0 and 100 is gray. Color index *a** (red–green axis), positive value is red, negative value is green. Color index *b** (yellow–blue axis), positive value is yellow, negative value is blue. Uppercase letters indicate significant differences between muscles (*p* < 0.05) and lowercase letters indicate significant differences between times (*p* < 0.05).

**Table 8 animals-14-00611-t008:** Glycolysis potential of different muscles of camel.

Item	Time	LT	PM	ST
Glucose (mmol/gprot)	1 h	3.91 ± 0.24 ^Aa^	2.81 ± 0.33 ^Ba^	3.80 ± 0.31 ^Aa^
	6 h	2.53 ± 0.46 ^Ab^	1.12 ± 0.12 ^Bb^	2.95 ± 0.66 ^Ab^
	24 h	2.03 ± 0.11 ^Bc^	0.79 ± 0.32 ^Cbc^	2.68 ± 0.39 ^Ab^
	72 h	1.56 ± 0.37 ^Acd^	0.46 ± 0.21 ^Bc^	1.96 ± 0.56 ^Ac^
	120 h	1.24 ± 0.27 ^Ad^	0.47 ± 0.25 ^Bc^	1.59 ± 0.27 ^Ac^
Muscle glycogen (mmol/gprot)	1 h	6.31 ± 0.58 ^Aa^	4.13 ± 0.21 ^Ba^	6.05 ± 0.19 ^Aa^
	6 h	3.67 ± 0.40 ^Ab^	2.51 ± 0.45 ^Bb^	3.67 ± 0.29 ^Ab^
	24 h	2.06 ± 0.46 ^Ac^	1.77 ± 0.32 ^Bc^	2.71 ± 0.32 ^Ac^
	72 h	1.56 ± 0.19 ^Ac^	1.43 ± 0.38 ^Ac^	1.64 ± 0.43 ^Ad^
	120 h	1.29 ± 0.01 ^Ac^	1.39 ± 0.07 ^Ac^	1.53 ± 0.39 ^Ad^
6PG (nmol/g prot)	1 h	2932.09 ± 206.46 ^Aa^	2562.29 ± 79.35 ^Ba^	2634.36 ± 103.03 ^Ba^
	6 h	1717.40 ± 15.45 ^Ab^	1008.98 ± 16.45 ^Bb^	1599.95 ± 30.01 ^Ab^
	24 h	638.41 ± 26.36 ^Bc^	624.24 ± 38.62 ^Bc^	787.25 ± 42.02 ^Ac^
	72 h	623.54 ± 25.19 ^Bc^	300.55 ± 6.62 ^Cd^	731.96 ± 42.90 ^Acd^
	120 h	590.83 ± 34.70 ^Ac^	289.08 ± 64.90 ^Cd^	641.79 ± 2.88 ^Ad^
Lactic acid (mmol/gprot)	1 h	0.84 ± 0.03 ^Ac^	0.41 ± 0.05 ^Cc^	0.57 ± 0.09 ^Bc^
	6 h	1.20 ± 0.01 ^Ab^	0.87 ± 0.02 ^Cb^	0.93 ± 0.03 ^Bb^
	24 h	1.29 ± 0.03 ^Aa^	0.90 ± 0.07 ^Cb^	1.09 ± 0.05 ^Ba^
	72 h	1.31 ± 0.04 ^Aa^	0.94 ± 0.12 ^Cb^	1.15 ± 0.07 ^Ba^
	120 h	1.33 ± 0.02 ^Aa^	1.05 ± 0.03 ^Ca^	1.17 ± 0.04 ^Ba^
GP (mmol/gprot)	1 h	27.14 ± 1.16 ^Aa^	19.41 ± 0.41 ^Ca^	24.41 ± 0.38 ^Ba^
	6 h	16.64 ± 0.81 ^Ab^	10.14 ± 0.91 ^Bb^	15.52 ± 0.59 ^Ab^
	24 h	10.74 ± 0.93 ^Ac^	7.27 ± 0.65 ^Bc^	11.26 ± 0.63 ^Ac^
	72 h	8.80 ± 0.38 ^Ad^	5.32 ± 0.76 ^Bd^	7.52 ± 0.85 ^Ad^
	120 h	7.57 ± 0.02 ^Ad^	5.35 ± 0.14 ^Cd^	6.35 ± 0.79 ^Bd^

Uppercase letters indicate significant differences between muscles (*p* < 0.05) and lowercase letters indicate significant differences between times (*p* < 0.05). gprot: g of protein.

## Data Availability

Data are contained within the article and Appendix A.

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
