# Peer review of "Effects of Muscle Type and Aging on Glycolysis and Physicochemical Quality Properties of *Bactrian camel* (*Camelus bactrianus*) Meat"

_animals, 2024, doi:10.3390/ani14040611_

Round 1
Reviewer 1 Report
Comments and Suggestions for Authors
OVERVIEW
This study aimed to examine the impact of muscle fiber type on glycolysis and the quality of Bactrian camel meat during various aging periods. The study's originality and novelty are significant, given the scarcity of research available on the specific meat aspects of these animals. However, it is important to note that there are certain concerns regarding the experimental design and statistical analysis employed in the study beside other minor remarks. Nevertheless, the subject matter of the paper aligns with the aims and scope of the journal, and the manuscript contains the main sections typically found in a scientific publication.
MAJOR COMMENTS
§ It is concerning that the authors did not account for the interaction between muscle type and aging time in their study. The experimental design should have employed a two-way ANOVA instead of a one-way ANOVA. This is because the study involves two independent factors: muscle type (with three levels) and aging time (with five levels), resulting in a 3 x 5 arrangement. To accurately analyze the data, the statistical analysis should follow this design. Addressing this critical issue is crucial before delving into the other details of the manuscript.
§ The tables require additional explanation and descriptive comments to enhance clarity and understanding.
§ The manuscript would greatly benefit from a thorough revision to address language usage, grammar, and style concerns. Improving these aspects will enhance the overall readability and professionalism of the paper.
MINOR COMMENTS
Title
The title would benefit from reconsideration to enhance clarity and readability. Here is a suggested modification:
"Effects of Muscle Type and Aging on Glycolysis and Physicochemical Quality Properties of Bactrian Camel (Camelus bactrianus) Meat”
Abstract
§ The abstract section needs to be revised to include the essential elements of a typical abstract. It should commence with an introductory sentence that provides context. Subsequently, the main objective of the study and the research problem should be stated clearly. Following that, the study design and major findings need to be presented. Finally, the abstract should conclude with a succinct statement based on the research outcomes. By adhering to this structure, the abstract will effectively summarize the key features of the manuscript, including its objective, research problem, study design, major findings, and a conclusive statement based on the research outcomes.
§ P1-L22: Which part of the Longissimus muscle was used in this study? Is it L. thoracis (LT), or L. lumborum (LL), or L. cervicis (LC), …etc
§ P1-L23: The number of used animals is not matching with that appeared in the M&M section, please check and make any necessary corrections.
Keywords
The word "ripening" does not appear in any part of the manuscript, except in this section. Therefore, it is recommended to replace it with a more appropriate term that accurately reflects the content of the manuscript.
Introduction
This section provides a solid introduction to the topic of the study and effectively justifies the reasons behind it. However, it is important to update the cited literature to ensure its relevance. For instance, an example of updated literature could be the study by Si (2022) published in the Foods Journal. Furthermore, it appears that this section may be overly extensive and may contain some parts that are not directly relevant to the study's context. It would be beneficial to review and streamline the section, focusing on the key points that directly support the study's objectives and research question. By doing so, the section will become more concise and maintain a stronger focus on the specific topic of investigation.
§ P1-L36- L38: The introductory part of this section appears to be unrelated to the topic of the manuscript and should be removed to maintain focus. Instead, it would be more appropriate to include a brief overview of the Bactrian camel and the specific breed used in the study. This will provide relevant background information and contextualize the research within the specific animal species being investigated. Including these details will enhance the coherence and relevance of the section to the study's subject matter.
§ P2-L57- L58: A reference is needed.
§ P2-L72- L73: The sentence needs revision for ease of understanding.
§ P3-L109- L114: The sentences are confusing, and the objective of the study is not clearly stated. Please revise them to improve clarity and understanding.
Methodology
The M & M section is well-written, providing an extensive explanation of the methodology used in the study. However, certain parts require revision to enhance detail and clarity. Additionally, the design and statistical analysis need to be reconsidered in terms of the main treatments, their levels, and whether the design should be a one-way or two-way ANOVA.
§ P3-L124: the number of the experimental animals was not as that mentioned in the Abstract section. Please, check and correct whenever possible.
§ P3-L125: Is it common practice to slaughter animals intended for meat consumption at this age? If not, what are the reasons behind slaughtering them at a later age, considering that most of the meat quality attributes are expected to be inferior at these later ages?
§ P3-L124-L125: The breed and sex of the experimental animals were not mentioned. Is Sonid the name of the area or the breed or both?
§ P3-L133-L134: The statement of approval by the respective department is already mentioned in lines 126-130, unless the latter refers to something different.
§ P3-L145: Please, define which part of the Longissimus muscle was used in this study. Moreover, the space between the12th and 13th ribs define the rib-eye which is only small part of the Longissimus muscle. Please, make sure of the exact part from which you have obtained the muscle sample.
§ P4-L151: It is preferable to avoid beginning a sentence with a numerical digit; instead, it is better to write the number in full.
§ P5-L205: It is preferable to avoid beginning a sentence with an abbreviation.
§ P5-L205 and L209: When was the exact time of measuring pH and color of meat? Were the samples for color measurement have experienced any form of freezing before performing the test?
§ P6-L241: As mentioned before, the experimental design and statistical analysis need reconsideration.
Results
The data were presented in 7 regular tables and 4 figures. The results were adequately described, but there were instances of extensive description and interpretation. This section should solely focus on presenting the data without any explanations or interpretations, which will be addressed in the Discussion section. Furthermore, some abbreviations were used in the tables without corresponding definitions in the table notes.
§ P7-L271: The muscle fiber proportion and area were existed in the table, but not the diameter as stated in the Table title. Please, revise and correct whenever needed.
§ P8-L314 and L325: Tables 5 and 6: units of measurement were missed. Please, add in place.
§ P8-L330: The L for lightness missed an asterisk (L*). It’s important to clarify whether CIE (L*, a*, b*) values or CIE Lab value were used. The presence or absence of the asterisk is an indication of mathematical differences between the two methods.
§ P9-L335: Table 7, be consistent in presenting the results. In this table, the position of aging periods and muscles was interchanged.
§ P9-L348: Table 8: what is the meaning of gpot? If it’s an abbreviation for g of protein, please make it clear or define it at the bottom of the table.
Discussion
The obtained results are appropriately interpreted and compared with other cited reviews.
§ P14-L471: Reference [26] is not for Kim, but Kadim as appeared in listed references. Please, check and correct if needed.
Conclusions
The conclusion requires reassessment as it overlooks some significant points. Restating the study's objective and highlighting the major limitations of the research are crucial elements of a conclusion. In this section, it is important to emphasize the significance of the findings rather than merely repeating them. I strongly encourage rewriting this section to ensure a robust conclusion that is grounded in the study outcomes.
References
Reasonable.
Comments on the Quality of English LanguageThe language used in writing this manuscript is generally clear and understandable. However, some grammatical and usage errors were observed. A comprehensive revision of the entire manuscript will result in a good paper.
Author Response
Hello Professor! We have made the appropriate changes to the issues you have raised and highlighted them in green. Below are the details of the changes.
- analytical methods and experimental design: for our interaction between ageing time and muscle fibre type on meat quality, we considered their direct correlation coefficients. We used Bivariate Correlation Analysis.In the results section, such as results section 3.1 and 3.2, a one-way ANOVA was used and we considered only the effect of ageing time on meat quality. The analysis of the results of the different muscle fibre types was also analysed in this section using one-way ANOVA, and the effect of ageing on the muscle fibre types was not done during the design of the experiment because the muscle fibres dissolve during ageing, which leads to a decrease in the shear force of the meat. Moreover, the experiment for myofibre detection requires that the meat samples must be dehydrated within 1h after slaughter and stored frozen at -80℃. After aging, meat samples cannot be tested for muscle fibre, and we also found that even at -80°C, if meat samples are left for more than one year, the probability of muscle fibre detection will not be successful. Prior to designing the experiment, we also learned from a literature review that the transformation of myofibrils is a lengthy process, and by adding probiotics to the diet, changes in myofibrillar composition were only detected at 3 months. The maximum aging time in this study was 7 days, which is not sufficient to support the completion of myofibre changes in meat samples. Furthermore, the numerous studies on myofibres cited in literature 16, 40, 42, etc. do not have the effect of post-slaughter ageing time on the composition of myofibre types.
- The title was changed as you suggested.
- The abstract was rewritten to clarify the purpose of the study, methodology, results and conclusions.
- P1-L22: The study is on longissimus thoracis (LT) and the supplement has been completed.
- Amend the keyword "ripening" to read "aging".
- P1-L36: Modified by deleting the original text and replacing it with a description of the Sunit sheep.
- P2-L57-L58: references added.
- P2-L72-L73: Sentences were revised.
- P3-L109-L114: revised language and added research items.
- Corrections were made to the number of animals in the abstract.
- Already answered in 1.
- P3-L125: Camels live to be 50 years old, cattle live to be 15 years old, and camels develop more slowly than cattle. The common slaughter age for camels in China is 8-10 years. Specific reasons are also added in P3-L122-125.
- P3-L124-L125: Sunit is the name of the place as well as the name of the breed, complementing the sex and breed of the animal.
- P3-L133-L134: Duplicates have been deleted.
- P3-L145: Longissimus thoracis was used in this study, and amendments have been completed.
- P4-L151 and P5-L205: Changes have been made as required.
- P5-L205 and L209: the time of assaying indicators such as pH and colour was added to p232, and the operation of sample collection was added to L175-183. pH and colour were assayed immediately after the samples were aged to the appropriate time, and the glycolytic potential and activity of the key enzymes of glycolysis were stored in liquid nitrogen after aging to the appropriate time, and then stored in liquid nitrogen at -80°C after being transported back to the laboratory. The samples were stored in liquid nitrogen after aging for the appropriate time, and then returned to the laboratory and stored at -80°
- P6-L241: Explained in 1.
- P6-L271: The table has been corrected.
- P8-L314 and L325: Units of measurement supplemented by Tables 5 and 6.
- P8-L330: This study used L*, which has been corrected and an explanation of L*, a* and b* is given in the table notes.
- P9-L335: Modified as requested.
- P9-L348: gpot means g of protein and has been amended as requested.
- P14-L471: Errors corrected.
- Conclusions are rewritten to ensure that solid conclusions are drawn based on the findings of the study.
Reviewer 2 Report
Comments and Suggestions for Authors
A brief summary
The proposed study focuses on the investigation of muscle characteristics from camel contributing to overall meat quality using three main muscles: the longissimus thoracis, psoas major, and semitendinosus. Functional properties accounted for included pH, tenderness, and color. Muscle fiber type was included to better capture what is happening in muscles. In the end, investigators reported varying outcomes between muscles. Overall, this basic information is useful and necessary to improve and better understand how muscles originating from camel can be marketable and provide a positive eating experience. The timing of this data is appropriate and much needed considering the lack in data involving meat quality of camel.
General concept comments.
Areas of Weakness
The information provided by this study is necessary and valid. However, weaknesses of this study include a lack in clarity on the end goal of this study. Another words, in investigating the “quality” of muscles from camel, quality should be defined as it can vary across confounding factors involving consumers. It would be helpful to include sensory data from consumers likely to consume meat products from camel. It would be helpful to compare meat originating from camel to beef and/or lamb as briefly mentioned in the introduction. There is a lack of details mentioned in the materials and methods that creates weakness of the methodologies. For instance, size of the LT muscle is not clear. It is assumed sample collections were done postmortem, but a time post slaughter is not clear. Units in testing should be included in addition to the mentioning of sample collection done in duplicate or triplicate (pH, color, tenderness). Tables are unclear of the units the values represent. Additionally, table titles do not describe type of statistic values represent (ex: basic means versus least significant differences).
Review
Overall, this type of study can provide invaluable data for the meat industry involving camel meat. This is valuable and necessary. However, completeness of this study is lacking. Details are missing in the materials and methods preventing an accurate replication of study.
Specific comments
|
LINE # |
COMMENT |
|
53 |
Clarify camel consumed in the world? North Africa? Etc. Who does the total amount include? |
|
72 |
Explaining/Identify quality. What does the consumer look for in camel meat? What does superior meat products mean? |
|
76 |
The inclusion of sensory data could be helpful. |
|
79-81 |
This information is contradictory. The inclusion of previous studies indicating higher pH meat results in more tender meat. |
|
110-113 |
A brief explanation on why the three muscles were chosen to investigate would be helpful. |
|
131 |
A brief mention of typical natural grasses consumed by camels will be useful. |
|
140 |
Replace the word “killed” with slaughter. |
|
144-147 |
Clarity on when samples were taken postmortem of slaughter. Were samples taken on the slaughter floor during or after the slaughtering process? Under refrigerated conditions? |
|
145 |
The LT muscle removed is confusing. Size, weight, thickness, etc is not clarified. |
|
207 |
Replace taken with recorded. |
|
209 |
Replace color difference with objective color values |
|
212 |
Number of readings is not clarified. Readings done in duplicate? Triplicate? Then averaged? |
|
220 |
Replace Shearing Force with Shear Force (throughout manuscript) |
|
224 |
Replace the word “trimmed” with cut. |
|
224 |
Was absorbent paper used to dry the surface of samples? Should be included. |
|
228 |
Units shear force was measured must be mentioned. (i.e.newtons, kg, lbs, etc.) |
|
256 |
Figure 1 should include names of muscles matching images left to right. |
|
|
Titles of tables are unclear or what values represent. Title should include: Least significant differences of … Ex: Table 4. Least significant differences of values representing pH values of muscles of camel postmortem. |
|
305 |
Replace Shearing Force with Shear Force |
|
314 |
Table 5 should include units of shear force values |
|
325 |
Title should include the percent sign to indicate values represent percentages. Ex; Least significant differences in cooking losses (%) during aging of different muscles originating from camel postmortem. |
|
336 |
Footnote should include what L*,a*, and b* values represent. |
|
348 |
Table 8. Decimals should align. It would be helpful to provide all values to the same hundredths place. Ex: 1.2 vs. 1.29. |
|
519 |
The inclusion of the current data compared to other meats, such as beef and/or lamb would be helpful. The introduction mentions beef and lamb. |
Comments on the Quality of English Language
Minor editing of English language required
Author Response
Hello Professor! We have amended the issues you raised appropriately and highlighted them in yellow. Below are the details of the changes.
- L-53:This figure is a United Nations survey of camel consumption worldwide, mainly in North African countries, and the total amount includes all countries globally. Changes have been made in the text.
- L-72:Consumers are looking for camel meat that is well tenderised and brightly coloured. The super meat in the text was an error in translation and has been corrected to high quality meat.
- L-76:Does sensory data here mean sensory indicators such as the odour of the camel's meat? We didn't find any articles related to the study of sensory metrics in camel aging, and I'm assuming you're talking about the impact of adding specific research data and metrics. We have modified it according to this.
- L-79-81:A faster rate of pH drop results in tougher, less tender meat. We have made changes in the article.
- L-110-113:added to the text why these three sites were studied and why this age was studied.
- L-131:Natural grasses used by camels were supplemented.
- L-131:Replace the word “killed” with slaughter.
- L-144-147:We collected the samples immediately after the workers stripped the skin and hair, the slaughter was carried out indoors at a temperature of 25-27° The samples were treated differently according to the experimental requirements. Samples for myofibre type testing were frozen and preserved, and samples for glycolysis-related indexes were frozen and preserved after aging for different periods of time so that they could be transported back to the laboratory for testing. The samples for the meat quality assay were tested in the slaughterhouse after aging to the corresponding time. In the article, I describe in more detail the collection and processing of the samples related to the different experiments.
- L-145:The weight of the samples was described, and the samples were collected in a square shape to allow for experimental manipulation, with no specific requirements.
- L-207:Replace taken with recorded.
- L-209:Replace color difference with objective color values.
- L-212:The reading methodology was supplemented.
- L-220:Replace Shearing Force with Shear Force (throughout manuscript).
- L-224:Replace the word “trimmed” with cut.
- L-224:bsorbent paper was used, which was modified in the text.
- L-228:The table is supplemented with units for shear and cooking losses.
- L-256:Figure 1 supplements the pictures with the corresponding muscle parts, the title of the table is unclear because the pictures were misplaced, and the different muscle fibre types corresponding to I, IIa and IIb are labelled in the new pictures.
- L-305:Replace Shearing Force with Shear Force.
- L-325:Units labelled with cooking losses.
- L-336:Explanations of L*, a* and b* have been added to the footnotes.
- L-384:The data in the full text is aligned to the second decimal place,EX:00.
- L-519:Comparison of camel meat with other meats added.
Round 2
Reviewer 1 Report
Comments and Suggestions for Authors
I would like to express my sincere appreciation to the authors for their commendable dedication in revising their manuscripts and addressing the raised concerns. Nearly all the points were taken into careful consideration, resulting in a significantly improved version of the manuscript that is highly acceptable.
It appears that one point may have been overlooked, which was as follows:
§ P5-L205 and L209: When was the exact time of measuring pH and color of meat? Did the samples for color measurement experience any form of freezing before performing the test?
Comments on the Quality of English LanguageThe language of the manuscript has shown significant improvement, but there are still some minor mistakes that could be rectified during the production process.
Author Response
Hello, Professor!
Thank you for your comment.
I have added p4 L-174-179 for the determination of pH, colour, cooking loss and shear time. No thawing was required for this experiment as the samples were placed at 4°C. The size or shape of the desired meat sample was added at p5 L234 and p6 L-239. Changes are therefore highlighted with yellow markers.